# Intelligent Platform Based on Smart PPE for Safety in Workplaces

**DOI:** 10.3390/s21144652

**Published:** 2021-07-07

**Authors:** Sergio Márquez-Sánchez, Israel Campero-Jurado, Jorge Herrera-Santos, Sara Rodríguez, Juan M. Corchado

**Affiliations:** 1BISITE Research Group, University of Salamanca, Calle Espejo s/n, Edificio Multiusos I+D+i, 37007 Salamanca, Spain; jorgehsmp@usal.es (J.H.-S.); srg@usal.es (S.R.); corchado@usal.es (J.M.C.); 2Air Institute, IoT Digital Innovation Hub (Spain), 37188 Salamanca, Spain; 3Department of Mathematics and Computer Science, Eindhoven University of Technology, 5600MB Eindhoven, The Netherlands; i.campero.jurado@tue.nl; 4Department of Electronics, Information and Communication, Faculty of Engineering, Osaka Institute of Technology, Osaka 535-8585, Japan; 5Faculty of Creative Technology & Heritage, Universiti Malaysia Kelantan, Locked Bag 01, Bachok, Kota Bharu 16300, Kelantan, Malaysia

**Keywords:** AIoT, anomaly detection, artificial intelligence, smart PPE, machine learning, Deeptech, edge computing

## Abstract

It is estimated that we spend one-third of our lives at work. It is therefore vital to adapt traditional equipment and systems used in the working environment to the new technological paradigm so that the industry is connected and, at the same time, workers are as safe and protected as possible. Thanks to Smart Personal Protective Equipment (PPE) and wearable technologies, information about the workers and their environment can be extracted to reduce the rate of accidents and occupational illness, leading to a significant improvement. This article proposes an architecture that employs three pieces of PPE: a helmet, a bracelet and a belt, which process the collected information using artificial intelligence (AI) techniques through edge computing. The proposed system guarantees the workers’ safety and integrity through the early prediction and notification of anomalies detected in their environment. Models such as convolutional neural networks, long short-term memory, Gaussian Models were joined by interpreting the information with a graph, where different heuristics were used to weight the outputs as a whole, where finally a support vector machine weighted the votes of the models with an area under the curve of 0.81.

## 1. Introduction and Motivation

There is a necessity to mitigate the high number of fatalities reported in recent years in the work environment. Different industries such as construction, mining or electricity have begun to invest in improving their employees’ safety by integrating new technologies or “smart technologies” in the work environment. These technologies are responsible for monitoring and protecting individuals in a work area. The integration of these systems aims to create an ecosystem of PPE to preserve the workers’ integrity. Such equipment must be adapted to the workers’ needs; providing protection without impeding them from performing their tasks as normal. This aspect is crucial for an effective application of the systems, as an uncomfortable or bulky device would discourage workers from its use. Therefore, the technology best suited to these requirements is wearable technology, which allows it to be worn as another accessory or garment. The International Labour Organisation (ILO) estimates that some 2.3 million persons around the world succumb to work-related accidents or diseases every year; this corresponds to over 6000 deaths every single day. Worldwide, there are around 340 million occupational accidents and 160 million victims of work-related illnesses annually, and the updates indicate an increase of accidents and ill health. However, statistics show that the injuries of workers may vary depending on their gender and nationality. First-world countries with more regulations and less dangerous jobs have a lower death rate compared with underdeveloped countries that also tend to not track workplace injuries or fatalities [1]. For example, in the United States, the incidence rate for total Occupational Safety and Health Administration (OSHA) recordable cases remained at 2.8 per 100 full-time workers and a total of 5333 occupational fatalities were recorded in 2019 according to the United States Bureau of Labor Statistics. This makes it necessary to research and develop technologies that replace or protect individuals at work. The impact of safety laws, safety programs regulations and occupational injury record-keeping regulations is of great importance. Effective rules and standards can significantly reduce accident and injury rates, consequently affecting business and the economy [2]. Different studies show the importance of state and federal agencies considering adopting workplace safety regulations [3] because of the impact these laws have on the frequency of reported workplace injuries and the consequent improvement of workers’ conditions [4].

Recent technological advances have revolutionised wearable technology, especially in 2002 when Bluetooth connectivity enabled wireless interaction between devices and has seen tremendous expansion over the last two decades [5,6]. With the market growing continuously, the number of such wearable applications and formats has been increasing steadily. Considering the availability and feasibility of wearable technologies and their ability to be an integral part of any individual’s daily life, they are the perfect candidate for the deployment of personal protective equipment in the work environment. The wearables on the market have adapted to the requirements of different niches. Understandably, not all of these adaptations were met with commercial success, either because the target population was not big enough or because of the existence of other more ergonomic solutions.

In recent years, the availability of wearable Internet of Things (IoT) devices has increased due to the cost reduction in technology production and making them more accessible to the public, provoking a great interest in studying this topic among the scientific community. This accessibility coupled with the IoT revolution allows us to gain access to some working environments’ complex mechanics and react quickly to imminent danger. IoT are networks of physical objects that implement technologies such as electronic circuits. Software, sensors, actuators, and network connectivity allow these objects to collect and exchange data for specific purposes [7]. Research has been carried out on equipment and auxiliary systems for detecting, warning and identifying risks, which can be integrated into different smart PPE. To this end, the different solutions that can be integrated in the device have been analysed, optimising the detection and prevention of risks, as well as the detection of the health conditions inherent to specific tasks in the workplace. All this is made possible thanks to the integration of sensors and actuators. Thus, an analysis has been made of the possibilities of adapting, developing and optimising existing technologies for: measuring anthropometric parameters, human activity recognition, real-time location systems and sensor networks. Thanks to the measurement of anthropometric parameters, the proposed bracelet is able to establish estimates of the worker’s condition in real-time, where a set of determined biometric measures is within pre-set thresholds. In addition, the architecture of the solution and the blocks into which it is divided have been clearly defined. We can find many examples of smart Personal Protective Equipment (PPE) developments that can monitor vital signs and the parameters of the industrial environment. In the paragraphs that follow, the different ways in which wearables can be integrated into workplaces are described, and it is analysed how ecosystems of connected devices can help protect the individual [8,9,10].

PPE is used to extend the individual’s capabilities and monitor their condition. For example, in professions such as fishing, warning and survival systems are integrated into protective clothing, including an artificial lung, flotation system and emergency light. The suit, which is highly visible in colour, alerts the rescue base station and gives the GPS position of the suit. In the case of firefighters, suits have been developed that integrate vital and environmental sensors, which aim to monitor the individual remotely and provide information to the individual about their environment and alert them of potential dangers via an HUD [11,12,13,14].

Smart PEE is a project inspired by the Ebola epidemic in Africa during 2013–2016 and which has been revitalised by the current pandemic. The project aims to redesign the personal protective equipment used by health workers and essential personnel during pandemics. The protective suit consists of a single protective piece that seeks to facilitate the use of the suit and reduce contact when the individual dresses or undresses. The suit is equipped with temperature, humidity, and air quality sensors inside the suit, and an integrated ventilation system that aims to keep the individual at a comfortable temperature so that they can get dressed and undressed. The suit is equipped with sensors for temperature, humidity and air quality inside the suit, and an integrated ventilation system that regularly keeps the individual at a comfortable temperature and takes care of the air quality [15,16,17].

These proposals include devices that are similar to those analysed and modelled in this research. In this context, if we analyse the suppliers of this type of product, we find worldwide companies (3M, Siteandfield, General Electric, Honeywell, etc.) whose positioning has been developed over decades and which offer classic protective equipment (helmets, masks, gloves, smartwatch, detectors, etc.) of high quality, at a competitive price and which try to incorporate applications that offer some advanced functionality. In recent years, newer companies (DAQRI, Human Condition Safety, Intellinium, Seebo) have also become involved in this market and are committed to providing solutions that integrate new technologies, such as Big Data and ICT solutions, to enhance industry 4.0 with a special focus on worker safety and the maintenance of facilities [18,19,20,21,22,23].

Systems capable of monitoring the state of workers and the environment are needed to ensure safe conditions and can also be integrated with other lines related to production, enterprise resource planning (ERP), etc. There are few examples of platforms that provide comprehensive support to workers, and those that we do find are very industry-specific. Some developments integrate virtual organisation technologies for the fusion of information from multi-domain data sources or neural networks, fuzzy logic, Bayesian networks, decision trees and other hybrid inference and artificial intelligence techniques. In this regard, our platform intends to meet the needs of any industry through a modular and adaptable design, having the ability to interoperate different individual protection devices, allowing for real-time visualisation and early detection of any anomaly using artificial intelligence. The platform incorporates a smart data system capable of providing intelligent responses, fusion algorithms and data mining processes of the different connected devices and a knowledge base that reduces the impact of anomalies.

Currently, there are platforms that use cloud architectures to process large amounts of data. Thanks to virtual agent organisations, the platform is modular and scalable, facilitating the incorporation of new vertical solutions integrated into the horizontal design of the city platform [24,25]. Likewise, with the use of technologies such as edge computing, it is possible to propose environments capable of processing information at the device level, carrying out pre-processing, eliminating noise as well as managing data more efficiently and rationally [26,27]. Similarly, the nodes can act autonomously, communicating with the platform on an ad hoc basis when their services are required. It also allows for the introduction of security layers between the platform and the nodes, which increases the reliability of the data. Thanks to the platform’s complementary capabilities (information extraction and integration capabilities, capacity to learn and to generate new rules using the acquired knowledge, high capacity to process raw data), the quality of the obtained information is improved. They allow the system to not only process data but also to understand it when performing tasks [28].

The trend is to integrate smart data capabilities into the platform of edge computing environments, capable of processing and combining data from different sources, such as ERP, manufacturing operation management (MOM), computerised maintenance management system (CMMS), supervisory control and data acquisition (SCADA), industrial Internet of Things (IIoT), vibrations, noise or incident reports in real-time. This capacity may be integrated with other technological innovations, such as cognitive systems, which are developed through the application of artificial intelligence to data, convolutional neural networks (CNN) and deep reinforcement learning (DRL), making it possible to build tools capable of controlling an enormous set of parameters related to processes and the environment. The aim is to gain exhaustive knowledge of the situation and incorporate intelligent mechanisms to optimise processes, maintenance and safety [29,30,31,32,33,34,35].

Regarding artificial intelligence models, the problem of maintenance and occupational safety in the industry has also been approached in different ways. Typically, the systems responsible for the safety of workers are designed to meet the specific needs of a company and tend to react to measured stimuli, provided that these reach a minimum threshold (action-reaction), which presents little capacity to adapt or modify behaviour in the face of new situations. On the other hand, AI systems in which learning mechanisms are introduced base their approach on a set of rules that, combined with stored knowledge obtained from solutions to known problems, try to predict the outcome of unpublished experiments. The use of neural networks (NN) [36], case-based reasoning (CBR) systems [37], deep learning (DL) [38] or hybrid neuro-symbolic algorithms [39] allow the system to determine whether a given situation poses a risk, depending on certain boundary conditions (the normal operation of a machine can pose a risk to a worker under certain circumstances and these do not necessarily have to be static).

Likewise, in different works focused on complex networks [40,41], it has been demonstrated that the analysis of information in the form of graphs allows a more complete structuring since non-linear information can be treated intuitively, which is why it has been proposed to transfer various machine learning techniques to complex networks, where links between pairs of nodes or relationships can be predicted based on previous information. For this purpose, heuristics well known in fields such as biology or computer science have been used.

From the above, it can be concluded that the use of technologies such as wearables, IoT, edge computing combined with wireless communications and artificial intelligence for decision making can bring about a significant evolution in safety and occupational risk prevention. In view of this, the implementation of an electronic system for the detection of anomalies in the workplace can be an essential support for the care of personnel. In this paper, we propose anomaly detection through three devices: a helmet [8], a belt [42], and a bracelet [43]. One of the main motivations of this work is the use of edge computing. This layer allows to process the information from the different devices, applying intelligent algorithms for the early detection of anomalies and pre-processing the data before being sent to the Cloud. Different artificial intelligence models are compared. Our previous work included the use of models such as convolutional neural networks, long short-term memory for the behaviour of industrial activities, Gaussian models for fall detection and alarm dispatching for fast responses. The models were linked through heuristics, where firstly it was proposed to interpret each model output as a graph. Models based on complex networks have shown great progress in recent years, so it was decided to interpret the information in that way, where finally, a support vector machine weighted the votes of the models and its validation was shown.

The remaining part of this research is organised as follows: Section 2 presents an overview of the related literature. Section 3 details the system design and the developed devices. Section 4 presents the data analysis as well as the union of the different models for the final answer. Finally, in the last section, we present some conclusive remarks and future lines of research.

The advantages and disadvantages of our paper with respect to other similar works have been included in Table 1, also showing the main technologies used in these works.

## 2. Platform Design

Currently, companies in the industrial sector focus their efforts on incorporating the advances included in the Industry 4.0 model to continue competing in an increasingly high-tech market. This proposal includes the development of a control and visualisation platform that integrates hybrid artificial intelligence algorithms to detect and prevent risks and accidents. As part of the development, different devices have also been developed that are connected to the platform for identifying risk situations so that the worker can be notified. The proposed system [22] is based on an architecture where there is a concentrator node consisting of a NVIDIA Jetson Nano, which is responsible for collecting the measurements received by the different devices in JSON format. Through communication with the Mosquitto server with MQTT communication protocol, the data will be sent to the remote server, reducing the sending rate to the essential data, thus deploying an edge computing system. The devices that have been developed are a helmet, belt and bracelet, which can be integrated into the platform to carry out the data analysis. As the data logger, we ideally use a NVIDIA Jetson Nano as an IoT edge device because it is a powerful low-power board, which allows carrying out intensive computations of algorithms based on machine learning. The described system can be seen in the following Figure 1 where there are three distinct layers. In the lower layer of devices, data collection and detection of individual alarms is carried out separately for each device. Subsequently, it communicates via WiFi with the intermediate layer in which data are collected from each one. In this layer, edge computing technology is used; all the system information is processed, applying intelligent algorithms to this layer for the early detection of anomalies and pre-processing the data before being sent to the Cloud in an orderly manner. Finally, the information is received in a Cloud environment for the visualisation of the data from the platform and the application of deep learning models to detect possible anomalies thanks to the training of the set of data ingested historically.

If we consider the hardware part within the different layers, simple rules have been implemented in each device’s firmware that alerts in real-time if any value has exceeded the limit value. On the other hand, in the concentrator node or gateway edge computing, advanced data processing is carried out, applying machine learning techniques and intelligent artificial intelligence algorithms. With them, we can detect if, based on all the information collected and historical data, an alert could be detected. In addition, we pre-process the data before sending it to the Cloud for visualisation. Therefore, concerning the devices and the Cloud server, this node acquires a fog computing node characteristic in the overall system.

These components are integrated with each other in order to provide monitoring of the operator’s environmental conditions. The information transmission component is an ESP32 module in each of them, which integrates WiFi technology in order to transmit the information collected from the components described above to a local server. The logical viewpoint of the electronic system collects information from the components responsible for monitoring the worker’s environment as well as the energy percentage of the lithium battery, as shown in Figure 2. After collecting the information, it is transmitted to the NVIDIA Jetson Nano via JSON. Likewise, with the aim of notifying the worker previously about the detected anomaly, threshold values are established to report the anomaly detected in the first instance. If the anomaly is confirmed, the information is retransmitted to the electronic system, being notified more frequently by the device. The power supply is provided in all the modules by a 3.7 V Lipo battery, connected to the power module. It should be noted that specific sensors integrated into the devices are repeated, as is the case of the Square Force-Sensitive Resistor (FSR), which, when placed in the helmet, allows us to detect whether we are wearing it and the impacts. On the other hand, the bracelet allows the operator to activate an alarm when it is pressed. In addition, the IMUs are repeated in all the devices to measure falls and impacts, using MPU6050 in the helmet and belt, and BMI160 in the bracelet. The derived advantages are that we will have more information for training and accident detection in the system. In addition, we will be able to have the functionality of reading and fall detection using the devices separately.

The Smart PPE helmet protects the operator from possible impacts while monitoring variables in their environment, such as light, humidity, temperature, atmospheric pressure, presence of gases, and air quality. At the same time, the smart PPE is to be bright enough to be seen by other workers, and the light source will provide a different vision to the operator [8]. The component used to supervise gas, harmful gases, pressure, temperature and humidity is the environmental sensor BME680. It is a micro-electromechanical system (MEMS) that integrates a volatile organic compounds (VOC) sensor, temperature sensor, humidity sensor and barometer. The sensor implemented for the monitoring of the level of brightness is the ALS-PT19 ambient light sensor. The sensor implemented for shock detection is a sensitive force resistor. The sensor responsible for detecting falls suffered by the worker is the MPU6050 module, an electronic component with six axes. The light source integrated into the helmet is a multicolour NeoPixel Adafruit LED strip, which will light up in a different colour depending on the alarm or the environment’s state. Figure 3 shows the helmet with the electronics installed and the display panel for the different parameters it measures.

The Smart PPE bracelet consists of electronic components welded to a flexible protoboard. These are ESP32 with an LCD display, buzzer, LED strip and monitoring sensor composed by IMU, temperature sensor, pulse sensor and panic button. The objective is to favour the interaction and monitoring of the user and the environment. For human activity recognition (HAR), a BMI160 inertial sensor (IMU) from Bosch (Stuttgart, Germany) provides precise acceleration and angular rate (gyroscopic) measurement. We measured the user’s body temperature through a Thermocouple Type-K Glass Braid Insulated Stainless Steel Tip from Adafruit (New York, NY, USA), which are best used for measuring surface temperatures. To measure the pulse rate, we used photodiode sensors, which have two light emitters and two light receivers and can measure the heartbeat, causing the reflected light to vary at each instant; this makes it possible to estimate the measurement of the pulses per minute. Figure 4 shows the bracelet with the panel and different sensor indicators.

The belt is composed of electronics integrated on a single board for fall detection, noise level and also makes it possible for the worker to send a warning if an anomaly has been detected or if they are in danger. As with the helmet and bracelet, it communicates with the NVIDIA Jetson Nano for sensor data processing and performing alarm detection on the device itself. To detect falls, an MPU6050 accelerometer was used, which contains an accelerometer and a MEMS gyroscope. To measure high noise levels, a KY-038 sound sensor was used, which is a transducer that converts the sound waves into electrical signals, incorporating a microphone together with an LM393 comparator, which allows reading both an analogue and a digital value. In addition, an accessible panic button is located outside of the electrical enclosure to alert the operator of a possible accident and override a false alarm [42]. Figure 5 shows the belt with the panel and different sensor indicators.

Table 2 shows the technical specifications of each of the electronic components selected for the device.

This set of devices forms an environment that, once deployed, monitors the position and status of machines and individuals, communication between employees, representation and analysis of data to prevent accidents and a real-time representation of the work environment through a remotely accessed digital model, Figure 6.

## 3. Data Analysis and Modelling

As mentioned in the introduction, this section integrates the different proposals that have been made to ensure the safety of workers in hostile environments, which can also be implemented in medical areas in the future. Our work unites three previous proposals: the intelligent multisensor helmet, the belt that integrates data aligned by the Naive Bayes model, and a bracelet that works with a hybrid model of a long short-term memory neural network and a Gaussian mixture model.

The ultimate goal is to have a complete platform with well-defined safety equipment for monitoring the strategic areas of the body, to reduce response times to accidents or problems that may occur during work that involves risk. That is, we will propose a way to handle the results of the previous devices with their respective AI/ML models (currently the method of ensemble models AdaBoost and Stacking ensemble ML are well known), where our proposal is to perform the stacking of ML models and deep models by means of the representation of the information as a complex graph. This allows us to handle information with non-linear behaviour in a natural way, where, in the end, we are obtaining a voting of the information as any other ensemble model would do.

### 3.1. Helmet

The smart helmet described in [8] is a piece of security equipment in which a comparison of different machine learning models was carried out to find an adaptable one for the analysis of human activity behaviour. The training dataset consisted of 11,755 samples and 12 different scenarios, the use of a deep convolutional neural network (ConvNet/CNN) is proposed for the detection of possible occupational risks, the CNN had an accuracy of 92.05% in cross-validation. This work was proposed to work with 2-D ConvNet where the instances of the data acquired were five features:Brightness,Variation in *X*, *Y* and *Z* axis,Force sensitive resistor,Temperature, humidity, pressure,Air quality.

See Figure 7 for the results of the valuation that were restructured for this article, which show the different sensors that were analysed for the possible situations in which the worker was subjected to certain conditions. Table 3 shows different possible circumstances ranging from having an environment conducive to work (such as good air quality, sufficient lighting) to having structural or physical risk (such as detecting falls through the CNN, lack of lighting, environment compromised by harmful gases). In the present work, the objective was to reduce reaction times in the event of an accident or mishap. For the smart helmet developed in our previous research, in which 12 different classes of events were classified for prompt action by personnel in charge of ensuring the safety of workers in industrial areas, the final performance of the CNN was 92.05% on average for all classes (represented with different colours). Cross-validation resulted in 20% for the CNN for the smart helmet.

### 3.2. Smart Bracelet with Platform

The smart bracelet is a device designed to be worn on the operator’s hand [43]. The inputs of the model are the body temperature of the user, their heart rate and a variable that indicates the status of the bracelet, that is to say, the optimal condition that is given by the battery or physical coupling. A label among four situations in a real working environment can be expected, such as a heart attack alert through agitation or falls, etc.

Problems recorded in the environment

Heart attack and irregular heartbeat,Extreme temperature changes leading to a heat stroke,Unhealthy temperature for the worker,Slips, trips and falls,Blows to the worker’s hand,Reporting an accident,

The bracelet uses two models. The first is the Gaussian mixture models focused solely on the analysis of the vital signs of the worker or user, as well as the status of the bracelet. The second model is the LSTM neural network, used to analyse the human activity behaviour, see Figure 8 where it shows the classification results of a GMM for four different classes, with an average performance in real-time of 78.75%, and Figure 9 where the results obtained for the GMM and LSTM are shown, respectively. The confusion matrix LSTM indicates the other model selected for the developed bracelet as GMM did not have a sufficient level of generalisation by itself. It can be seen that the best class performance was 96.14% for the WalkingScaling class, 89.2% for Falls and 77.06% for Carrying, which is the most difficult class to predict in our dataset.

### 3.3. Smart Belt Design by Naive Bayes Classifier

The purpose of the belt is [42] very similar to the two previous ones; however, in each device, there is a variation from the perspective of the accelerometers due to the position of the body where they are located, which is why no information is discarded because it is a complement. For the belt, the following labels were obtained, where it is worth mentioning that each device was subjected to a Fisher statistical analysis to verify the non-overlapping of classes.

Problems recorded in the environment and its label

Low Battery 1,Z-axis difference greater than low value 2,Z-axis difference greater than average value 3,Z-axis difference greater than high value 4,High decibels 5,Panic button on 6,Low battery and difference on Z axis greater than low value 7,Low battery and difference on Z axis greater than average value 8,Low battery and difference on Z axis greater than high value 9,Low battery and high decibels 10,Low battery panic button activated 11,Difference on Z axis greater than low value and high decibels 12,Z-axis difference greater than mean value and high decibels 13,Z-axis difference greater than high value and high decibels 14,Z-axis difference greater than low value and emergency button activated 15,Z-axis difference greater than mean value and emergency button activated 16,Z-axis difference greater than high value and emergency button activated 17,Panic button activated and high decibels 18,Z-axis difference greater than low value and high decibels and low battery 19,Z-axis difference greater than mean value and high decibels and low battery 20,Z-axis difference greater than high value and high decibels and low battery 21.

### 3.4. Integration

To integrate all the information, it is important to recapitulate that the smart helmet has 12 possible labels, the bracelet has a total of 7 distributed in 2 different models, the GMM and the LSTM, and finally the belt has 12. Our work proposes to see each of the different outputs as nodes in a network, where information modelling has significantly better results than in traditional models.

That is why our approach is based on not only analysing the 40 possible labels but also being prepared for possible contingencies where one of the devices is inactive or it is necessary to reduce the noise/risk of prediction/classification, for which we propose the use of heuristics and machine learning to filter the final information in the proposed network, see Figure 10. This diagram represents how the labels (final output) of each model are used together to create a graph that allows us to have each label represented as a node. The reader may have noticed that we are talking about model ensembles with the difference that our boosting is given by an independent dataset on each model. Our ultimate goal is to integrate everything into a decision making process that in turn generalises a vote.

The graph created with the different labels can be seen in Figure 11. This image represents the relationship that exists between the labels when analysing the three devices in parallel; as mentioned, we want to reduce the risk of misclassification by finding the relationship between the output labels of each model, i.e., the input of the analysis by heuristics is the output of the models of the electronic devices. For the network generated with the different outputs of the four integrated models, each colour represents belonging to a certain model, for example, the 3 orange nodes represent the output of the LSTM (3 classes), the 4 blue nodes the 4 of the GMM, the 12 green nodes to the CNN and the rest to the Naive Bayes. This is how to achieve the joint responses of all the models.

#### Procedure

Machine learning (ML) is the study of computer algorithms that improve automatically through experience. This means that it is possible to make predictions or approximations on the basis of information with similar characteristics. In ML, we have two big categories, unsupervised and supervised learning. In the current project, we work with supervised learning, which is a technique for deducing a function from training data. Training data consists of pairs of objects (usually vectors): one component of the pair is the input data and the other is the desired results. For that we need to create a training dataset.

By creating a dataset for validation, we choose an unknown part of the multisensory network for the model in order to evaluate its performance. Regularly, in ML, we train the models with features that represent the nature of the data, but what kind of information can be used in graphs? One option is heuristics. Heuristics allow designing a score based on network topology.

The Adamic–Adar index is a measure introduced in 2003 by Lada Adamic and Eytan Adar to predict links in a social network. The Adamic–Adar of a pair of nodes (u) and (v) is defined as in Equation (Equation 1):

(1)$$\sum_{w\in \Gamma (u)\cap \Gamma (v)}\frac{1}{\log|\Gamma (w)|}$$ where (Γ(u)) denotes the set of neighbours of (u). That is why *w* is the intersection neighbours between both nodes.

Another heuristic is the Jaccard coefficient, which measures the similarity between finite sample sets, and is defined as the size of the intersection divided by the size of the union of the sample sets. The Jaccard coefficient of nodes (u) and (v) is defined as in Equation (Equation 2), where | means the absolute value:(2)$$\frac{\left|\Gamma (u)\cap \Gamma (v)\right|}{\left|\Gamma (u)\cup \Gamma (v)\right|}\;\;\;\;\left(10\right)$$
where (Γ(u)) denotes the set of neighbours of (u).

Given a fixed network, is it possible to predict how the label network is going to look in the future? The Soundarajan–Hopcroft index indicates the count of common neighbours between two nodes, summed together with the count of common neighbours that belong to the same community as the two nodes. For two nodes (u) and (v), this function computes the number of common neighbours and a bonus one for each common neighbour belonging to the same community as (u) and (v). Mathematically, Equation (Equation 3):(3)$$\sum_{w\in \Gamma (u)\cap \Gamma (v)}\frac{f(w)}{\left|\Gamma (w)\right|}\;\;\;\;\left(11\right)$$
where (f(w)) equals 1 if (w) belongs to the same community as (u) and (v) or 0 otherwise, and (Γ(u)) denotes the set of neighbours of (u).

The dependent variables (outputs) for the datasets will be created since they will indicate whether or not there is a relationship between each set of output labels. The heuristics have a very close approximation to the real relationships, which can be verified through the evaluation of the real data against the data obtained by each heuristic.

Once we have the heuristics we can use them as features to train our model and then use that information to try to predict links between the new network when the set of devices is used in real-time. *X* represents the input and can be provided as a list: each of the internal lists corresponds to the features of one node pair. *y* is the list of values to predict. *X* will be prepared by combining the heuristics that have been used.

Two common steps are involved in machine learning algorithms to learn how to combine heuristics for optimising predictions:Training: show features + value to predict;Using/Validating: try to predict value from features.

The general process is described in [40], where our objective is to find the relationship between the tags of the 3 devices and send a tag number 41, which allows us to validate the correct operation of the entire security platform. In summary, what is done is the following (example directed to link prediction):Starting from a graph, we choose a set of heuristics available in the stat-of-the-art for complex networks (the most common ones are: Jaccard coefficient, Hub Promoted, Adamic Adar, etc.).Referring to the Jaccard Coefficient, this allows us to establish a coefficient for a specific node (defined as similarity based on the neighbourhood of the node), this value will become a characteristic for the input vector that will be used for the final modelling.The process is repeated by assigning a value by a heuristic to each node pair combination in the network, where at the end we will have an input vector for each node pair with a target that is identified with 0 or 1, depending on whether or not there is a link between each node pair.Finally, an ML model is applied to the generated data where what we are doing is weighting the response of the different artificial intelligence models represented as a complex network to reduce the classification noise in industrial and work environments. The entire process is summarised in Figure 12.

Based on the above, we made a comparison between a Gaussian Naive Bayes, random forest, support vector machine (SVM) and a linear regression, where the result is quite favourable, see Figure 13. This allows us to see that there is a strong relationship when any of the labels warn the user of a danger in their environment or they are alerted of their vital signs, which likewise, allows the integration of the three devices that have been developed.

Here, the entire process is summarised. We have four models integrated into three different devices. There is an analogy in the field of artificial intelligence that a single model is not as good as several models together looking for something specific (analogy of the elephant and the blind men). That is why we decided to include a decision-making process for different wearables to get a picture of what is really going on. Several of these models have independent variables in common but are not linearly related. This allows us to make a decision based on votes as an AdaBoost model, where our Boosting comes from the independence of each dataset. The complex network allows all outputs to be represented as nodes for easy analysis and once this is done we can now use heuristics to determine if a situation is actually happening or not.

## 4. Discussion and Conclusions

The increasing presence of new technologies in the workplace has become more than evident in the last two decades, revolutionising work environments and increasing productivity. In the near future, we will witness an increase in the application of these technologies to ensure the well-being and health of employees, adapting the form of smart PPE and becoming more accessible to the public once their application and use become popular. The popularisation of smart PPE will be crucial in metallurgy, mining, fishing, commercial and construction due to their high-risk rate and high lethality. The development of smart PPE will be driven in tandem by advancing artificial intelligence, which will enable systems to become more adaptive and intelligent and meet the widespread demand for safer workplaces.

The implementation of IoT and wearable technology has revolutionised the field of smart PPE. Workers are individually monitored by the sensors implemented in wearables. The information can be sent to a central node that monitors and provides personalised responses to different individuals on the basis of the data received from them. These benefits have already been recognised by different industries experimenting with these technologies and their applications in their work environments. The platforms that companies currently have in place focus on production and productivity aspects. One component to reduce these causes is the implementation of protective measures, especially where it is not possible to determine standards to ensure the integrity of the individual. As a result, it is important to use PPE, such as belts, helmets and other devices that protect the physical integrity of the operator. Furthermore, with the use of edge computing, we can implement this solution in areas with a lack of connectivity and relatively cheaply with the consequent impact on reducing casualties or accidents in any country.

Legal requirements force companies to make significant investments in protective equipment, an area in which they strive to stand out from among their competitors and to comply with regulations at a reasonable cost. For this reason, the use of technologies such as IoT, Big Data and Cloud Computing in protective equipment is of high interest to companies. The BeSafe 2.0 platform implements next-generation technologies in these fields (AIoT, smart data, advanced machine learning, predictive algorithms and advanced support decision systems).

An important goal for the future is to patent or register the hardware, algorithms, and derived services etc., as intellectual property, so that the investment that is being made is safeguarded and the technology can be extrapolated to other potential markets, exploiting the results through multiple business models.

Finally, it is necessary to continue developing new proposals and protective equipment for different areas of application, adapting the technology to meet their security needs. Workplace accidents are a very important problem for worker’s care due to various factors that represent a risk to them.

Likewise, by implementing artificial intelligence techniques in the hardware, it is possible to improve the predictive capacity of the electronic system, placing it as an essential step for the improvement of equipment by implementing emerging technologies in the hardware of the PPE. In addition, the implementation of 5G technology has been contemplated, which has the great advantage of supporting a higher bandwidth, which will translate into higher download speeds [23].

## Figures and Tables

**Figure 1 sensors-21-04652-f001:**
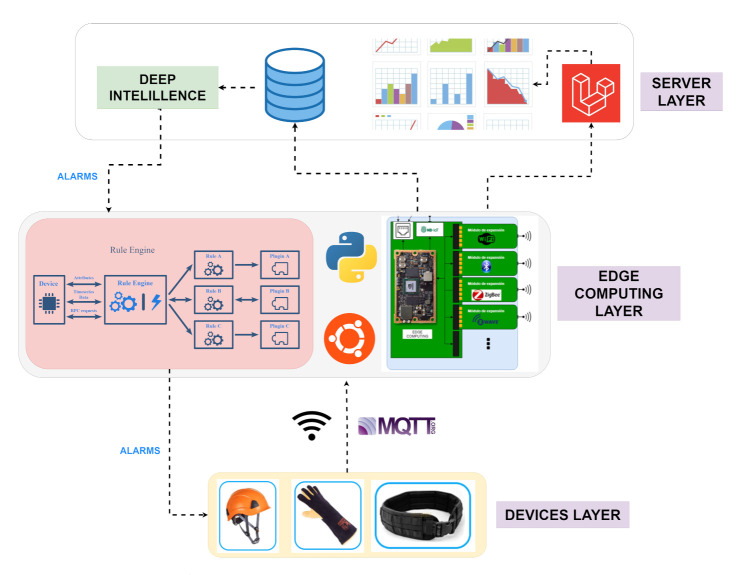
System architecture.

**Figure 2 sensors-21-04652-f002:**
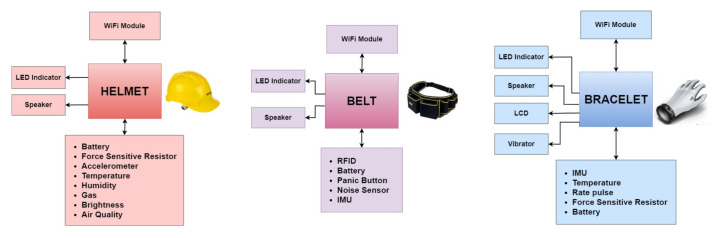
Devices diagram [8,42,43].

**Figure 3 sensors-21-04652-f003:**
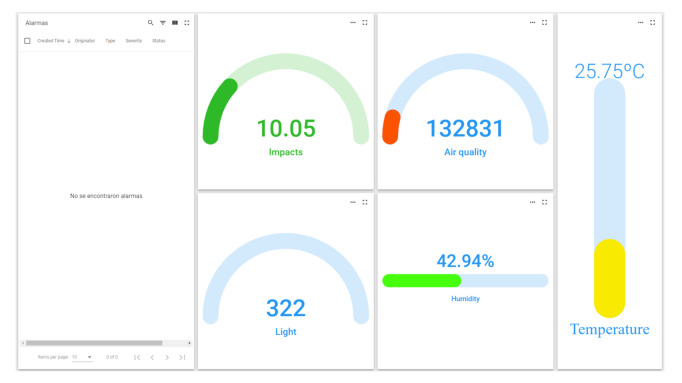
Platform helmet alarm panel [8].

**Figure 4 sensors-21-04652-f004:**
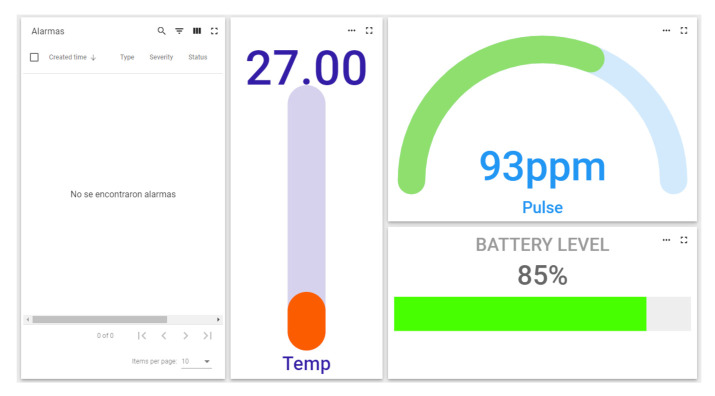
Platform bracelet alarm panel [43].

**Figure 5 sensors-21-04652-f005:**
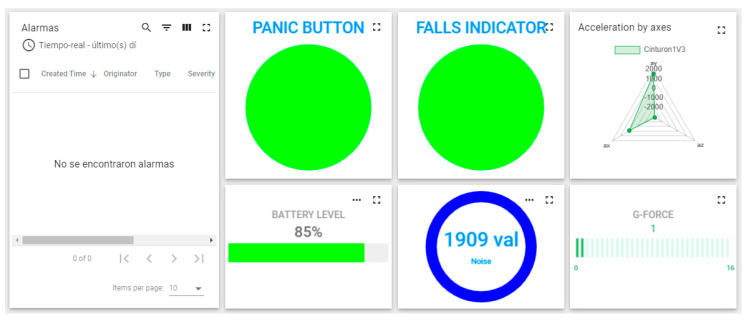
Platform belt alarm panel [42].

**Figure 6 sensors-21-04652-f006:**
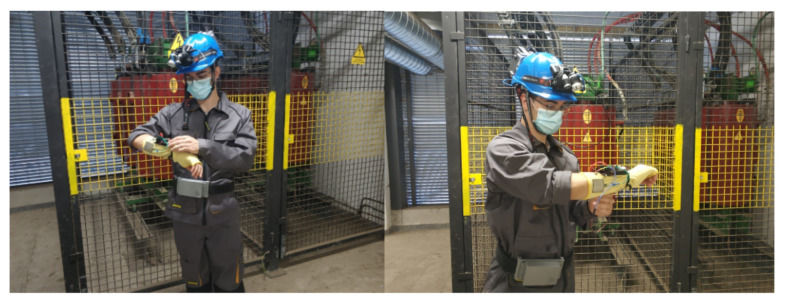
The configuration of the bracelet, helmet and belt.

**Figure 7 sensors-21-04652-f007:**
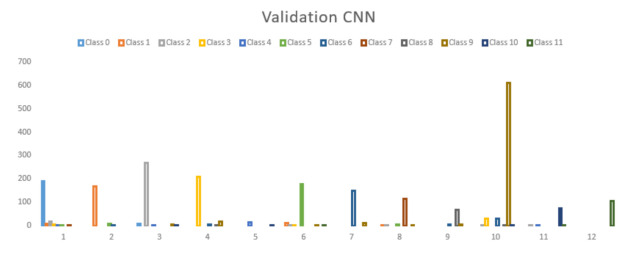
Smart helmet developed in our previous research [43].

**Figure 8 sensors-21-04652-f008:**
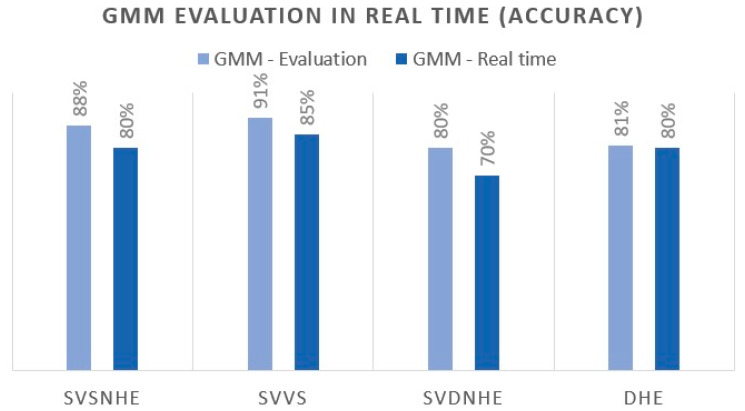
Results of GMM in real-time, where the definition of the labels are: worker with stable vital signs and non-hostile environment, which we will define as SVSNHE, worker with smooth variation in vital signs, SVVS, worker with vital signs in dangerous and non-hostile environments, defined here as SVDNHE, and worker in danger due to hostile environment, DHE. Bracelet developed for human behavioural analysis.

**Figure 9 sensors-21-04652-f009:**
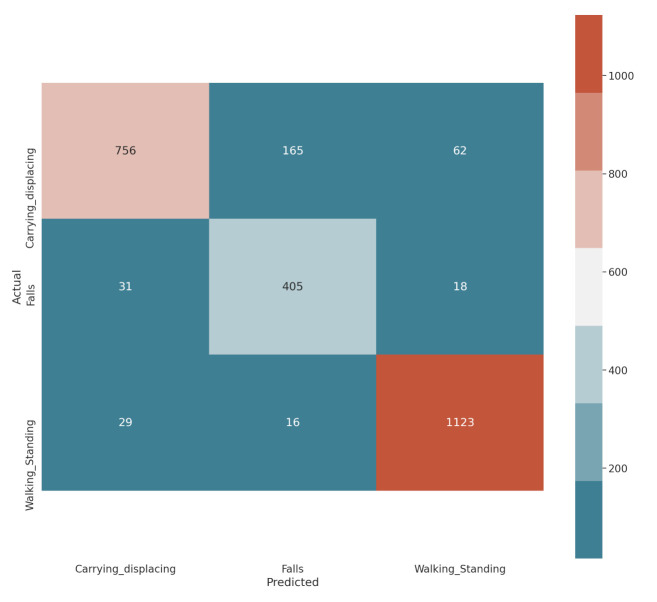
Confusion matrix LSTM.

**Figure 10 sensors-21-04652-f010:**
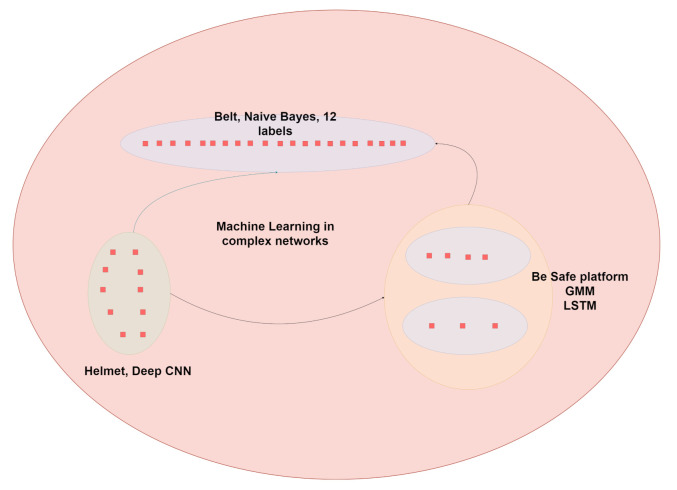
Final interpretation of devices through heuristics and machine learning applied to complex networks.

**Figure 11 sensors-21-04652-f011:**
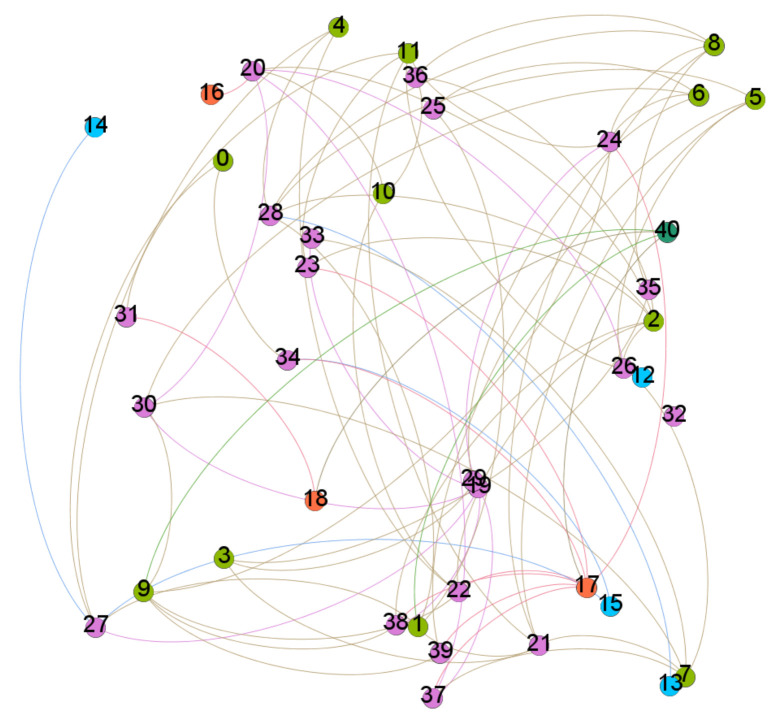
Image obtained by plotting the network in the Gephi software.

**Figure 12 sensors-21-04652-f012:**
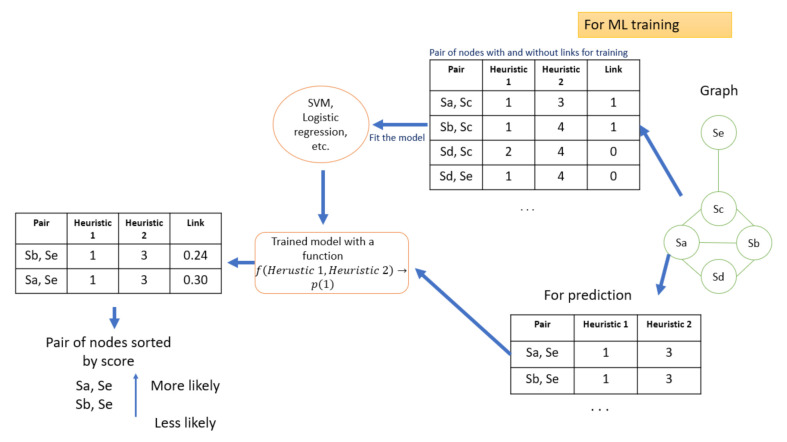
Explanation of the process of including ML for joint decision making over an entire network. The flow of the diagram is from right to left.

**Figure 13 sensors-21-04652-f013:**
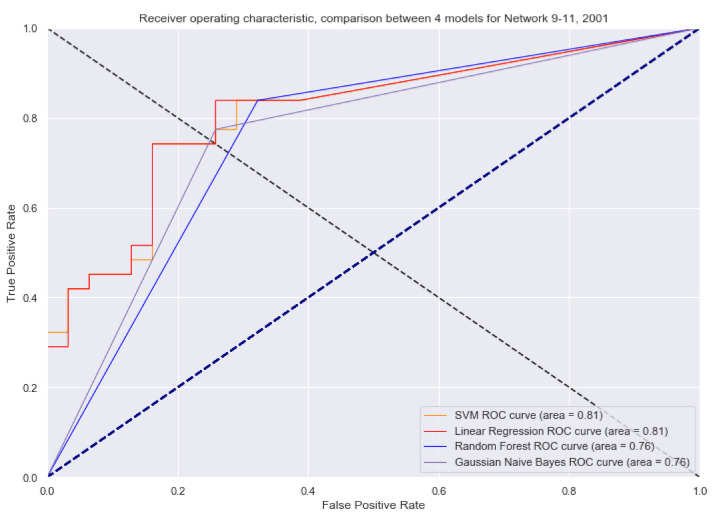
Receiver operating characteristic (ROC) curves in the analysis of different models for the integration of the complete multiplatform.

**Table 1 sensors-21-04652-t001:** Proposals related to similar technologies and platforms.

Bibliografy	Technologies Included	Advantages and Disadvantages	Novelty of the Proposal
Heras, Stella, et al. (2012)		This work is focused on the use of multi-agent systems and cloud computing, so our platform provides additional functionality.	It presents the role of argumentation in the next generation of agreement technologies and their use in cloud computing environments.
Shawish, Ahmed, and Maria Salama. (2014)	Cloud computing, Distributed systems, Multi-agent systems, Virtual organisations Social Computing	It classifies the Cloud’s deployment and service models, providing a complete description of the Cloud services vendors.	It provides a comprehensive overview of the Cloud’s anatomy, definition, characteristic, effects, architecture, and core technology.
Chamoso, Pablo, et al. (2019)		Improves interaction between users and ensures a fast and secure operation.	A multi-agent system is the base of the developed software; MAS is an essential and commonly used tool in social computing.
Chamoso, Pablo, et al. (2016)		The system provides a web application to manage all the review processes for power lines management.	This work is focused on the periodic review of transmission towers (TT) to avoid important risks, such as step and touch potentials, for humans.
Riverola, Florentino Fdez, et al. (2000)	Hybrid neuro-symbolic systems, Case-based reasoning (CBR), Artificial Neural Networks (NN) Agent Virtual organisation, Deep Learning	Classifications of these systems, paying particular attention to each subsystem’s distinctive features that make up the hybrid models.	This work is a general review of hybrid neuro-symbolic artificial intelligence systems, focusing on those composed of artificial intelligence.
Van Den Oord, Aäron, et al. (2013)		Deep content-based focus on music.	This work is aimed at a platform based on music recommendation through deep learning.
Chen, Zhen-Yao, et al. (2017)		A hybrid of genetic algorithm and artificial immune system (HGAI) algorithm.	Evolutionary algorithm-based radial basis function neural network training for industrial personal computer sales forecasting.
Puig Ramírez, Joaquim. (2010)		The reliable detection and anticipation of performance deviations via monitoring the production and product-related process, diagnostic of possible causes and predicting the time of occurrence.	Asset optimisation and predictive maintenance in discrete manufacturing industry.
Mobley, R. Keith. (2002)	Predictive maintenance, Machine Train Monitoring, Industrial organisation, Production control, Supervision	The system provides maintenance methods in manufacturing or production plans.	It is a review of the methods and methodologies for carrying out predictive maintenance in industries.
Sittón, Inés, et al. (2017)		It is aimed at the recognition and extraction of unstructured data patterns from IoT sensors.	Pattern extraction for the design of predictive models in Industry 4.0
Shin, D. et al. (2016)		Use of edge computing, followed by several case studies, ranging from cloud offloading to smart home and city.	It presents several challenges and opportunities in the field of edge computing.
Shi, Weisong and Schahram Dustdar. (2016)	Edge computing, Real-time monitoring, Internet of Things (IoT), Protective System Smart cities and home	The success of the Internet of Things and rich cloud services have helped create edge computing.	This work is aimed at understanding edge computing technology and its multitude of applications to propose environments capable of processing information at the device level.
Satyanarayanan, Mahadev. (2017)		Industry investment and research interest in edge computing.	It presents several research and opportunities in the field of edge computing.
**Bibliography**	**Technologies Included**	**Advantages and Disadvantages**	**Novelty of the Proposal**
Sánchez, Sergio Márquez, et al. (2020)		The paper is before work that is supported by the current platform work based on ROS.	This work is based on edge computing Driven Smart Personal Protective System Deployed on NVIDIA Jetson and Integrated with ROS.
Sun, Xiang, and Nirwan Ansari. (2016)		A hierarchical fog computing architecture in each fog node to provide flexible IoT services while maintaining user privacy.	Mobile edge computing for the Internet of Things.
Podgorski, Daniel, et al. (2017)		A new conceptual framework for dynamic Occupational Safety and Health (OSH) management in Smart Working Environments (SWE)	A proposed framework is based on a new paradigm of OSH risk management consisting of real-time risk assessment and the capacity to monitor the risk level of each worker individually.
Boyes, Hugh, et al. (2018)	Ambient intelligence (AI), Industrial Internet of Things (IIoT), Smart working environment (SWE), Occupational safety management, Personal Protective Equipment (PPE), Cyber-physical systems, Real-time risk assessment,	It is focused on analyses related to partial IoT taxonomies.	It develops an analysis framework for IIoT that can be used to enumerate and characterise IIoT devices.
Sánchez, Sergio Márquez, (2019)		They seek to improve the health and safety of work sectors where there is a high risk of an accident.	Solutions made available by industry 4.0 to prevent hazards with a wireless model consists of the design of different innovative PPE.
Chae, Hye Seon, et al. (2017)		Smart personal protection equipment uses various biometric information from the combination of devices to allow the wearer to voluntarily recognize the danger	The research and development of the rural smart personalisation equipment for preventing farming and disaster prevention

**Table 2 sensors-21-04652-t002:** Components technical specifications.

Device	Component	Characteristics	Description
**Helmet**	ALS-PT19 Ambient light sensor	- Supply Voltage: Vcc −0.5∼6.0 V - Vce = 5 V, Ev = 1000 Lx - Color Temperature = 6500 K	The ALS-PT19 is a low cost ambient light sensor, consisting of phototransistor in miniature SMD.
	MPU6050	- Supply Voltage, VDD −0.5 V to +6 V - Acceleration (Any Axis, unpowered) 10,000 g for 0.2 ms - Gyroscope Features: FSR of ±250, ±500, ±1000, and 2000${}^{\circ }$/s - Accelerometer Features: FSR ±2 g, ±4, ±8 and ±16 g - Nonlinearity (typ.) (A): 0.5% (G): 0.2% - Sensitivity Scale Factor: (G.typ) 131, 65.5, 32.8, 16.4 LSB (${}^{\circ }$/s) (A.typ) 16.384 LSB/g, 8.192 LSB/g, 4.096 LSB/g, 2.048 LSB/g - Sensitivity Scale Factor Tolerance: (G) ±3% FSC: Full Scale Range	The MPU6050 module contains a three-axis gyroscope with which we can measure angular velocity and a 3-axis accelerometer with which we measure the X, Y and Z components of the acceleration, the accelerometer works on the piezo electric principle, it also has a temperature sensor.
	NeoPixel Adafruit LED strip	- Supply Voltage: Vcc +6.0∼+7.0 V - Low voltage output current: 18.5 mA and 10 mA (min) - Operation Frequency: 800 KHz	Neopixel stick 8 × 5050 RGBW LEDs ∼3000 K
	Square Force-Sensitive Resistor (FSR)	- Actuation Force ∼0.2 N min - Force Sensitivity Range: ∼0.2 N–20 N - Force Repeatability Single Part +/− 2% - Force Repeatability Part to Part +/− 6% (Single Batch)	FSRs are sensors that allow you to detect physical pressure, squeezing and weight.
	BME680	- Digital interface I^2^C (up to 3.4 MHz) and SPI (3 and 4 wire, up to 10 MHz) - Supply voltage: VDD: 1.71 V to 3.6 V - VDDIO: 1.2 V to 3.6 V - Operating range −40–+85 °C, 0–100% r.H., 300–1100 hPa	The BME680 is a digital 4-in-1 sensor with gas, humidity, pressure and temperature measurement based on proven sensing principles.
**Belt**	MPU6050	- Supply Voltage, VDD −0.5 V to +6 V - Acceleration (Any Axis, unpowered) 10,000 g for 0.2 ms - Gyroscope Features: FSR of ±250, ±500, ±1000, and 2000${}^{\circ }$/s - Accelerometer Features: FSR ±2 g, ±4 g, ±8 g and ±16 g - Nonlinearity (typ.) (A): 0.5% (G): 0.2% - Sensitivity Scale Factor: (G.typ) 131, 65.5, 32.8, 16.4 LSB (${}^{\circ }$/s) (A.typ) 16.384 LSB/g, 8.192 LSB/g, 4.096 LSB/g, 2.048 LSB/g - Sensitivity Scale Factor Tolerance: (G) ±3% FSC: Full Scale Range	The MPU6050 module contains a three-axis gyroscope with which we can measure angular velocity and a 3-axis accelerometer with which we measure the X, Y and Z components of the acceleration, the accelerometer works on the piezo-electric principle, it also has a temperature sensor.
	KY-038 sensor	Analogue Signal, VDD: 3.3 V	Microphone sound sensor module
**Bracelet**	Thermocouple Type-K	- Precision: ±1 °C - Output range: −6 to 20 mV	Glass braid insulated stainless steel tip, which can be used in high temperature.
	Heart Rate Monitor Sensor	- Input Voltage (Vin): 3.3–6 V (5V recommended) - Output Voltage: 0 - Vin (Analogue), 0/ Vin (Digital) - Operating current: <10 mA	It is based on PPG techniques, to detect blood volume changing in the microvascular bed of tissues.
	BMI160 Inertial sensor (IMU)	- Sensitivity (typ.) Acc. ±2 g: 16,384, ±4 g: 8192, ±8 g: 4096, ±16 g: 2048 LSB/g - Sensitivity (typ.) Gyro. ±125${}^{\circ }$/s: 262.4, ±250${}^{\circ }$/s: 131.2, ±500${}^{\circ }$/s: 65.6 LSB/${}^{\circ }$/s - TCS (typ.) (A): ±0.03%/K (G): ±0.02%/K - Nonlinearity (typ.) (A): 0.5 %FS (G): 0.1 %FS - Offset (typ.) (A): ±40 mg (G): ±3${}^{\circ }$/s - TCO (typ.) (A): ±1.0 mg/K (G): 0.05${}^{\circ }$/s/K	It is an inertial measurement unit (IMU) consisting of a state-of-art 3 axis, low-g accelerometer and a low power 3 axis gyroscope.
	Square Force-Sensitive Resistor (FSR)	- Actuation Force ∼0.2 N min - Force Sensitivity Range: ∼0.2 N–20 N - Force Repeatability Single Part +/− 2% - Force Repeatability Part to Part +/− 6% (Single Batch)	FSRs are sensors that allow you to detect physical pressure, squeezing and weight.

**Table 3 sensors-21-04652-t003:** Table showing the restructured labels of the work focused on the intelligent case, one of the three components used to model them through complex networks.

Label	Meaning in the Model
0	Good for health air (AQI from 0 to 50) with sufficient illumination in the working environment.
1	Moderate air quality (AQI of 51 to 100) with slight variation in temperature and humidity.
2	Harmful air to health for sensitive groups (AQI 101–150) with moderate variation in temperature and humidity.
3	Harmful air to health (AQI 151 to 200) with considerable variation in temperature and humidity
4	Very harmful air to health (AQI 201 to 300) with high variation in temperature and humidity.
5	Hazardous air (AQI greater than 300) with atypical variation in temperature and humidity.
6	Lack of illumination and variation equivalent to a fall in axes.
7	Lack of illumination and variation equivalent to a fall in axes and considerable force exerted on the helmet.
8	Atypical variation on the detected axes and moderate force detected on the FSR.
9	Illumination problems, air quality and sudden variation in axes.
10	Very high force exerted on the FSR.
11	Variation in axes with illumination problems.
12	Outliers on the 5 sensors

## Data Availability

Not applicable.

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
