# Peer review of "Intelligent Platform Based on Smart PPE for Safety in Workplaces"

_sensors, 2021, doi:10.3390/s21144652_

Round 1

Reviewer 1 Report

The work has the scope of “interoperate different individual protection devices, allowing for real-time visualization and early detection of any anomaly using Artificial Intelligence” . It describes the implementation of a platform for monitoring the individual that leverages on a wide set of sensors and, above all, the use of artificial intelligence techniques using local devices defined as edge computing. The work is neatly organised in 5 paragraphs. The introductory part presents a substantial generic enumeration of the evolutionary scenario of IoT systems. The second paragraph, The state of the art, describes some scenarios in which monitoring the individual is useful by relegating the final lines to short examples of artificial intelligence; The third part describes the HW instrumentation used including sensors and platform, declaring protocols and referring the reader to the previous works done by the same team. The fourth paragraph aims to describe the analytical and context modeling approach used in order to interpolate data obtained from the different physical devices and, therefore, how the ML was applied. The conclusions envisage the possibility of stimulating the development of such solutions by providing for their patenting.

The topic covered by the article is interesting, however there are some critical issues related to the innovation made with respect to previous works that could be highlighted more providing, for instance, more information about the advantage of using interpolation of all equipment data. Furthermore, the proposed architecture could be conceived with a vision more aimed at 5G infrastructures and Multi-access Edge Computing (MEC, Sun, Xiang and Nirwan Ansari. "EdgeIoT: Mobile edge computing for the Internet of Things." IEEE Communications Magazine 54.12 (2016): 22-29.)

Section 1 (Introductions and motivations) and section 2 (state of the art) could be improved by trying to include more specific information on the subject matter and, in particular, on how artificial intelligence is supporting the overcoming of specific problems in this area. In addition, it would be recommended to reduce the overall length of the sections.

In Section 3 (Platform Design), the description of the proposed architecture is not easy to understand and could be improved by indicating, for example, where the hardware is located and, moreover, by specifying better the server solution used. Figure 3 is reused from the team's previous work (see Fig. 6 and Fig. 3 by Campero-Jurado, Israel, et al. "Smart Helmet 5.0 for industrial internet of things using artificial intelligence." Sensors 20.21 (2020): 6241 .) Section 4 could provide more information to the reader to allow a better understanding of the choices made by the authors for data analysis and modelling and, therefore, facilitate understanding of the results proposed in Figure 12. Some typo have been found:

  • Line 40, “the availability of wearable devices has increased due to the reduction in technology production, making them more accessible to the public”: the sentence seems to be discord with the meaning of the period. Maybe the sentence should be “the availability of wearable devices has increased due to the cost reduction in technology production”;
  • Line 46: use “warning,” instead of “warning”;
  • Line 79: “Our previous work” would require a citation to be inserted;
  • Line 100: Use “acquired” instead of “acuquired”;
  • Line 133: use “humidity,” instead of “humidity”;
  • Line 139: use “warning,” instead of “warning”;
  • Line 152: remove comma from “ to enhance industry 4.0,”;
  • Line 189: Missing reference;
  • Line 195: “personal circumstances or of work”, use “personal or work circumstances” instead;
  • Line 277: use “representation,” instead of “representation”;
  • Line 275: wrong indentation;
  • Line 404: missing reference;

Some sentences are difficult to read and could be enhanced:

  • Line 162-166: the sentence is difficult to read;
  • Line 297: The sentence is hard to understand;
  • Line 397-399: it's no clear. which X and which y are considered.

Author Response

REVIEWER 1

The work has the scope of “interoperate different individual protection devices, allowing for real-time visualization and early detection of any anomaly using Artificial Intelligence” . It describes the implementation of a platform for monitoring the individual that leverages on a wide set of sensors and, above all, the use of artificial intelligence techniques using local devices defined as edge computing. The work is neatly organised in 5 paragraphs. The introductory part presents a substantial generic enumeration of the evolutionary scenario of IoT systems. The second paragraph, The state of the art, describes some scenarios in which monitoring the individual is useful by relegating the final lines to short examples of artificial intelligence; The third part describes the HW instrumentation used including sensors and platform, declaring protocols and referring the reader to the previous works done by the same team. The fourth paragraph aims to describe the analytical and context modeling approach used in order to interpolate data obtained from the different physical devices and, therefore, how the ML was applied. The conclusions envisage the possibility of stimulating the development of such solutions by providing for their patenting.

We are so thankful for your comments and reviews. We have modified and generated all the content based on your comments. We believe that the paper will be much more structured, and the research done will be more visible.

The topic covered by the article is interesting, however there are some critical issues related to the innovation made with respect to previous works that could be highlighted more providing, for instance, more information about the advantage of using interpolation of all equipment data. Furthermore, the proposed architecture could be conceived with a vision more aimed at 5G infrastructures and Multi-access Edge Computing (MEC, Sun, Xiang and Nirwan Ansari. "EdgeIoT: Mobile edge computing for the Internet of Things." IEEE Communications Magazine 54.12 (2016): 22-29.)

Thank you very much for the technological vision offered. We have considered an Edge Computing architecture for data processing because these are sensitive depending on the company. Filtering of the data is performed to send to the cloud only those that are of interest. The incorporation of 5G technology has been considered future work as we believe it could bring significant value to the architecture and send data with low latency and high speed.

In addition, a comparative table has been created showing the advantages and improvements offered by our work concerning other similar ones, also showing the main technologies used in these works.

Section 1 (Introductions and motivations) and section 2 (state of the art) could be improved by trying to include more specific information on the subject matter and, in particular, on how artificial intelligence is supporting the overcoming of specific problems in this area. In addition, it would be recommended to reduce the overall length of the sections.

Introduction and state of the art was compared against our current work to make it relevant. The advantages and disadvantages of our paper respecting to other similar ones have been included in Table 1. Also, we included a paragraph expressing the novelties of our work concerning others.

In Section 3 (Platform Design), the description of the proposed architecture is not easy to understand and could be improved by indicating, for example, where the hardware is located and, moreover, by specifying better the server solution used. 

Figrue 2 has been included as diagram describing the hardware architecture has been incorporated. Table 2 with the technical specifications of each of the electronic components selected for the device and the different anomalies we can detect have been reflected in section 4. We have already added more information to clarify our ideas and explanation about the data analysis part as well as the modeling. To do so, we add Figure 11.

Figure 3 is reused from the team's previous work (see Fig. 6 and Fig. 3 by Campero-Jurado, Israel, et al. "Smart Helmet 5.0 for industrial internet of things using artificial intelligence." Sensors 20.21 (2020): 6241 .)

As noted by other reviewers, the figure has been retained and the paper in which the device was worked on has been cited.

Section 4 could provide more information to the reader to allow a better understanding of the choices made by the authors for data analysis and modelling and, therefore, facilitate understanding of the results proposed in Figure 12. 

 Thank you very much for your comments. We have already added more information to clarify our ideas and explanation about the data analysis part as well as the modeling. To do so, we add Figure 11.

Some typo have been found:

  • Line 40, “the availability of wearable devices has increased due to the reduction in technology production, making them more accessible to the public”: the sentence seems to be discord with the meaning of the period. Maybe the sentence should be “the availability of wearable devices has increased due to the cost reduction in technology production”;
  • Line 46: use “warning,” instead of “warning”;
  • Line 79: “Our previous work” would require a citation to be inserted;
  • Line 100: Use “acquired” instead of “acuquired”;
  • Line 133: use “humidity,” instead of “humidity”;
  • Line 139: use “warning,” instead of “warning”;
  • Line 152: remove comma from “ to enhance industry 4.0,”;
  • Line 189: Missing reference;
  • Line 195: “personal circumstances or of work”, use “personal or work circumstances” instead;
  • Line 277: use “representation,” instead of “representation”;
  • Line 275: wrong indentation;
  • Line 404: missing reference;

Some sentences are difficult to read and could be enhanced:

  • Line 162-166: the sentence is difficult to read;
  • Line 297: The sentence is hard to understand;

Thank you very much for your review Dr., I have changed the structure of this sentence to make it more understandable.

  • Line 397-399: it's no clear. which X and which y are considered.

Thank you for your comment, we have already added a Figure 12 which we believe clarifies the process of how the dependent and independent variables are created to use Machine Learning in Complex Networks.

A revision of the language has been carried out on the whole paper, we detected some errors that we have already corrected. So, the references that were not cited have been included in the text and referenced them correctly.

Reviewer 2 Report

The paper really needs to be rewritten.  The sections are a mix of opinion and references to things like "the proposed platform" or "the bracelet" prior to any description of the platform or its components.  The graphs and figures are difficult to read and lack context or description of what is being shown.  The platform section lacks a clear diagram or description of the hardware and software, and uses terms like "scenarios" and "circumstances" and "situations."  Some text seems like it was lifted from a proposal.  There is description of what models are being used (GMM, LSTM, etc.) without showing how these provide any insight into safety. It is not clear what is significant about the scenarios, labels, etc. (for instance the list at the end of 4.3) or how they relate to situational awareness or safety.  

It is a shame because a paper describing how a smart helmet, bracelet, and belt could work together to detect specific safety-specific conditions would be of great interest, and I suspect that the authors have done some very good work using AI/ML to understand how the sensor values might predict fatigue.  But the paper is confusing and the results and methods presented are not well described, nor are any of the diagrams or graphs clear in what they are depicting (or why the reader should care).  A very very promising project but the paper fails to communicate the project or the progress made.

Author Response

We are so thankful for your comments and reviews. We have modified and generated all the content based on your comments. We believe that the paper will be much more structured, and the research done will be more visible.

The paper really needs to be rewritten.  The sections are a mix of opinion and references to things like "the proposed platform" or "the bracelet" prior to any description of the platform or its components. 

Thank you very much for your feedback. I have added a correct explanation to each diagram and image to clarify our ideas.

Introduction and state of the art was compared against our current work to make it relevant. A comparative table has been created “Table 1” showing the advantages and improvements offered by our work concerning other similar ones, also showing the main technologies used in these works. Also, we included a paragraph expressing the novelties of our work concerning others.

Figure 2 has been included as diagram describing the hardware architecture has been incorporated. Table 2 with the technical specifications of each of the electronic components selected for the device and the different anomalies we can detect have been reflected in section 4. We have already added more information to clarify our ideas and explanation about the data analysis part as well as the modeling. To do so, we add Figure 11.

The graphs and figures are difficult to read and lack context or description of what is being shown. 

Following the reviewer's recommendation, the quality of some images has been improved and they have been better explained in the text. . We have already added more information to clarify our ideas and explanation about the data analysis part as well as the modeling. To do so, we add Figure 11.

The platform section lacks a clear diagram or description of the hardware and software, and uses terms like "scenarios" and "circumstances" and "situations." 

A diagram describing the hardware architecture has been incorporated and the different anomalies we can detect have been reflected in Table 3.

Some text seems like it was lifted from a proposal.  There is description of what models are being used (GMM, LSTM, etc.) without showing how these provide any insight into safety. It is not clear what is significant about the scenarios, labels, etc. (for instance the list at the end of 4.3) or how they relate to situational awareness or safety. 

I thank you for your very wise observation.

We have added the information specifying the different situations a worker can go through and how we use ML in Complex Networks to create an ensemble model:

‘’ I summarise the whole process. We have 4 models integrated in 3 different devices, there is an analogy in the field of artificial intelligence that a single model is not as good as several models together looking for something specific (analogy of the elephant and the blind men). That is why we decided to include a decision-making process for different wearables to get a picture of what is really going on. Several of these models have independent variables in common but not linearly related. This allows us to make a decision based on votes as an AdaBoost model, where our Boosting comes from the independence of each dataset. The Complex Network allows all outputs to be represented as nodes for easy analysis and once this is done we can now use heuristics to determine if a situation is actually happening or not.’’

It is a shame because a paper describing how a smart helmet, bracelet, and belt could work together to detect specific safety-specific conditions would be of great interest, and I suspect that the authors have done some very good work using AI/ML to understand how the sensor values might predict fatigue.  But the paper is confusing and the results and methods presented are not well described, nor are any of the diagrams or graphs clear in what they are depicting (or why the reader should care).  A very very promising project but the paper fails to communicate the project or the progress made.

Thank you very much for your comments. We have already added more information to clarify our ideas and explanation about the data analysis part as well as the modeling. To do so, we add Figure 11.

A revision of the language has been carried out on the whole paper, we detected some errors that we have already corrected. So, the references that were not cited have been included in the text and referenced them correctly.

Reviewer 3 Report

important help for them. The work is great, but the submitted paper is not good enough as a scientific paper in Journals.

  1. The literature review is too lengthy, many of referred works are not relevant with the theme of this paper. Giving a table with simple descriptions and focusing on the most important works in this field may be helpful. Also, the relationship between the referred works and your paper is not well shown. I cannot get the necessity to proposed the Section 2.
  2. Many sensors are used in the helmet, bracelet and belt. Only type number is given. No manufacturer, no sensing capacity, No key parameters. The references of their datasheet should be cited.
  3. Source of Fig.1 is not cited.
  4. MPU6050 is a force sensor (referred in Line 252) or inertial sensor (Line 261)?
  5. Both helmet and bracelet can provide inertial signals by using the same type sensor (MPU6050). Why and how to use them in the following data processing?
  6. Figure 3 had been published in a paper by the same authors (Ref.37), but no citation.
  7. Wearing all these devices seems not very compact. Will it influent the works?
  8. How to get the values in Figure 7? How to get the accuracy from these values? The results in Figure 7 is not very clear in the current state.
  9. Additional numbers in Eqs. 2&3. The meaning?
  10. Line 404. A very important number is missing.
  11. The conclusions section is not good. I cannot get the achievements and finding of authors by reading it.

Author Response

important help for them. The work is great, but the submitted paper is not good enough as a scientific paper in Journals.

  1. The literature review is too lengthy, many of referred works are not relevant with the theme of this paper. Giving a table with simple descriptions and focusing on the most important works in this field may be helpful. Also, the relationship between the referred works and your paper is not well shown. I cannot get the necessity to proposed the Section 2.

We are so thankful for your comments and reviews. We have modified and generated all the content based on your comments. We believe that the paper will be much more structured, and the research done will be more visible.

In addition, a comparative table has been created showing the advantages and improvements offered by our work concerning other similar ones, also showing the main technologies used in these works.

Introduction and state of the art was compared against our current work to make it relevant. The advantages and disadvantages of our paper respecting to other similar ones have been included in Table 1. Also, we included a paragraph expressing the novelties of our work concerning others.

  1. Many sensors are used in the helmet, bracelet and belt. Only type number is given. No manufacturer, no sensing capacity, No key parameters. The references of their datasheet should be cited.

Table 2 has been included with the technical specifications of each of the electronic components selected for the device.

  1. Source of Fig.1 is not cited.

A reference it was included

  1. MPU6050 is a force sensor (referred in Line 252) or inertial sensor (Line 261)?

We use inertial sensor with MPU6050 an a force resister sensor on other hand

  1. Both helmet and bracelet can provide inertial signals by using the same type sensor (MPU6050). Why and how to use them in the following data processing?

We use MPU6050 in both cases for measuring falls and impacts

  1. Figure 3 had been published in a paper by the same authors (Ref.37), but no citation.

A reference to a previous work has been added

  1. Wearing all these devices seems not very compact. Will it influent the works?

Dear, thank you for your comment, we have added a paragraph responding to this question in the conclusions and future work. Indeed the sensors already included in the prototypes we have created have a certain invasive level, however, in the industrial area it is essential to use protective equipment, that is why our purpose has always been to integrate sensors to the equipment used since ever.

 How to get the values in Figure 7? How to get the accuracy from these values? The results in Figure 7 is not very clear in the current state.

Thank you very much for your accurate observation, we have changed the figure where now you can clearly see that a color belongs to a respective class and this in turn presents more accuracy on it than the slight overlap with the remaining ones.

  1. Additional numbers in Eqs. 2&3. The meaning?

Thank you very much for your accurate observation, we have added the meaning of each of the variables in the equations: {where $(gamma(u))$ denotes the set of neighbors of $(u)$. Thus $w$ is the intersection of neighbors between the two nodes.}

  1. Line 404. A very important number is missing.

Thank you again, we have added the missing reference, an apology for our oversight.

  1. The conclusions section is not good. I cannot get the achievements and finding of authors by reading it.

It was included some information about the achievements of our work.

Round 2

Reviewer 1 Report

The work has been revised in different sections making some improvements and making clearer some parts that were previously very difficult to understand. However, given the initial state, the revised paper failed to fill the gaps previously expressed.

Moreover, the added Table 1, Table 2 as well as figure 13 are not easy to read.

Author Response

Dear Dr., using your first comments we thank you for your support. We improved again the part of the introduction where we included more specific concepts on how artificial intelligence is used to support problems in the industry.

The scope of the work has been reflected and explained in the introduction.

The aim is to reduce accidents based on the data found in publications and verified statistical studies. All ambiguous sentences have been modified in order to make the message much clearer, without any opportunity for individual assessment or misleading.

The system architecture diagram has been better explained. Therefore it has been reflected as a motivation within the work and has also been supported by different publications that reflect the effort in developed and developing countries.

Likewise, in section 3 the general architecture of the system was explained to indicate how the 3 models interact given the 3 devices, the figures were detailed where the dependent and dependent variables of each model are explained as well as their previous performance. Specifically, the following paragraph was added (as an introduction to the data modeling and analysis part):

“The ultimate goal is to have a complete platform with well-defined safety equipment for monitoring the strategic areas of the body, to reduce response times to accidents or problems that may occur during work that involves risk. That is, we will propose a way to handle the results of the previous devices with their respective AI/ML models, currently it is well known the method of ensemble models, AdaBoost, Stacking ensemble ML, where our proposal is to perform the stacking of ML models and Deep models by means of the representation of the information as a complex graph. This allows us to handle information with non-linear behavior in a natural way, where, in the end, we are obtaining a voting of the information as any other ensemble model would do.”

In addition to that, the article has been sent with a specialized translator to improve the language problems.

Moreover, the added Table 1, Table 2 as well as figure 13 are not easy to read.

We thank you for your valuable contributions, we have improved the quality of the images and changed the colours when necessary, some of them were even recreated.

Reviewer 2 Report

This is a major improvement and reflects thoughtful work by the authors.  I believe some of the motivating materials are still in need of improvement, and will cite three things that I think need to be addressed.

  1. Background facts are insufficiently established, which undermines the credibility of the paper.  First sentence says "higher" number of deaths.  Higher than what?  In what industries relevant to the proposed PPE?  
  2. More damaging is the false claim that "first world" countries have "less regulations and more precarious employment" - this is simply not true.  Many countries (particularly non-first world or "developing" countries) do not track workplace injuries or deaths, whereas more advanced countries have both much (much) stronger safety laws and much (much) more extensive reporting laws.  Comparing these numbers to countries that do not count, or have lax laws on reporting, makes no sense. Moreover, please do a few Google searches and you will find that 5,000 deaths in the U.S. is extremely low compared to estimated tens of thousands in many other countries that report such data.  Even if true, the conjecture based on a single number in a single country is unsupported. (A grammatical note, “precarious employment” would mean the job is uncertain, and could be lost, which is different from “dangerous jobs.” And this statement is also verifiably not true – the safety laws in more advanced countries make jobs much safer than the same jobs in developing countries, and the economic conditions in advanced economies rely on more sophisticated equipment rather than humans for the most dangerous of jobs, which are still done by humans in developing countries.)
  3. Figure 2 is difficult to understand.  The physical devices are at bottom (3 pieces) and top (servers) but what is the middle layer?  Do not these functions run on the belt?  Yet the diagram shows the belt talking to the middle layer with mqtt and alarms.   I suggest not mixing physical with logical unless you make the diagram much more clear.

For item 2, I would suggest that the importance of this work is that using ever-less-expensive edge computation and sensing the smart PPE can have a safety impact in developing countries as well as developed countries.  That is, you are proposing technology that will have a greater impact than just in already developed countries (thus impacting many more people).  This strengthens the motivation for the work.  You should be able to find multiple references about workplace safety and how developed economies have enacted laws that make employers spend more money to reduce workplace injuries (in contrast to developing countries without such laws, where companies do NOT implement costly safety controls and mechanisms).

Again I applaud the authors for the diligent work of improving the manuscript, but strongly advise fixing the three items above.

Author Response

This is a major improvement and reflects thoughtful work by the authors.  I believe some of the motivating materials are still in need of improvement, and will cite three things that I think need to be addressed.

Thank you very much for the notes made by the reviewer. An has been made to modify all the points mentioned.

  1. Background facts are insufficiently established, which undermines the credibility of the paper.  First sentence says "higher" number of deaths.  Higher than what?  In what industries relevant to the proposed PPE?  

The scope of the work has been reflected and explained in the introduction.

  1. More damaging is the false claim that "first world" countries have "less regulations and more precarious employment" - this is simply not true.  Many countries (particularly non-first world or "developing" countries) do not track workplace injuries or deaths, whereas more advanced countries have both much (much) stronger safety laws and much (much) more extensive reporting laws.  Comparing these numbers to countries that do not count, or have lax laws on reporting, makes no sense. Moreover, please do a few Google searches and you will find that 5,000 deaths in the U.S. is extremely low compared to estimated tens of thousands in many other countries that report such data.  Even if true, the conjecture based on a single number in a single country is unsupported. (A grammatical note, “precarious employment” would mean the job is uncertain, and could be lost, which is different from “dangerous jobs.” And this statement is also verifiably not true – the safety laws in more advanced countries make jobs much safer than the same jobs in developing countries, and the economic conditions in advanced economies rely on more sophisticated equipment rather than humans for the most dangerous of jobs, which are still done by humans in developing countries.)

The aim is to reduce accidents based on the data found in publications and verified statistical studies. All ambiguous sentences have been modified in order to make the message much clearer, without any opportunity for individual assessment or misleading.

  1. Figure 2 is difficult to understand.  The physical devices are at bottom (3 pieces) and top (servers) but what is the middle layer?  Do not these functions run on the belt?  Yet the diagram shows the belt talking to the middle layer with mqtt and alarms.   I suggest not mixing physical with logical unless you make the diagram much more clear.

The system architecture diagram has been better explained. As the reviewer indicates, there are three layers, each with very distinct functions.

In the lower layer of devices, data collection and detection of individual alarms is carried out separately for each device.

Subsequently, it communicates via Wi-Fi with the intermediate layer in which data is collected from each one. In this layer is used Edge Computing technology; all the system information is processed, applying intelligent algorithms to this layer for the early detection of anomalies and pre-processing the data before being sent to the Cloud in an orderly manner.

Finally, the information is received in a Cloud environment for the visualisation of the data from the platform and the application of Deep Learning models to detect possible anomalies thanks to the training of the set of data ingested historically.

For item 2, I would suggest that the importance of this work is that using ever-less-expensive edge computation and sensing the smart PPE can have a safety impact in developing countries as well as developed countries.  That is, you are proposing technology that will have a greater impact than just in already developed countries (thus impacting many more people).  This strengthens the motivation for the work.  You should be able to find multiple references about workplace safety and how developed economies have enacted laws that make employers spend more money to reduce workplace injuries (in contrast to developing countries without such laws, where companies do NOT implement costly safety controls and mechanisms).

I agree with the reviewer's suggestion. One of the advantages of Edge Computing is that it can be implemented in areas with a lack of connectivity and relatively cheaply. With the consequent impact on the reduction of casualties or accidents in any country.

Therefore it has been reflected as a motivation within the work and has also been supported by different publications that reflect the effort in developed and developing countries.

Again I applaud the authors for the diligent work of improving the manuscript, but strongly advise fixing the three items above.

Reviewer 3 Report

The revision has improved some parts of this paper, but it still cannot make it for publication, in my opinion.

  1. About Section 2. I still cannot get the necessity to proposed the Section 2. It is lengthy and not match the main theme of this paper. I think it is better to delete it and move some main points to Introduction.
  2. Same sensors are used in different positions. Why? How to simultaneously use them in the following data processing operation? What happens it only one of them is used?
  3. The meaning of Fig.8 is not clear? The information is not enough. Also, values in Fig.8 has been published before or not?
  4. Many of devices and results have been published before, I suggest the authors to reorganized the Introduction to show the real value of this paper: not proposing a new whole system, but making a promotion. It is necessary to show the new items and findings, not repeating the existing things. Which results support these new findings? How can we easily and clearly get them? They are all important for this paper.
  5. The information in Figs.12 and 14 is not well shown. What can we find from these figures?
  6. The Abstract and Conclusion should be written to show the new findings of this paper.

Author Response

We are so thankful for your comments and reviews. We have modified and generated all the content based on your comments. We believe that the paper will be much more structured, and the research done will be more visible.

The revision has improved some parts of this paper, but it still cannot make it for publication, in my opinion.

  1. About Section 2. I still cannot get the necessity to proposed the Section 2. It is lengthy and not match the main theme of this paper. I think it is better to delete it and move some main points to Introduction.

I thank you for your comment, We have restructured section 1 and 2 of the paper, removing the less relevant content and arranging all the information in a clearer way.

  1. Same sensors are used in different positions. Why? How to simultaneously use them in the following data processing operation? What happens it only one of them is used?

Some specific sensors are integrated into the devices are repeated:

In the case of the Square Force-Sensitive Resistor (FSR), which, when placed in the helmet, allows us to detect whether we are wearing it and the impacts. On the other hand, the bracelet allows the operator to activate an alarm when pressed.

The IMUs are repeated in all the devices to measure falls and impacts, using MPU6050 in the helmet and belt, and BMI160 in the bracelet. The derived advantages are that we will have more information for training and accident detection in the system. In addition, we will be able to have the functionality of reading and fall detection using the devices separately.

  1. The meaning of Fig.8 is not clear? The information is not enough. Also, values in Fig.8 has been published before or not?

I thank you for your comment, the information in Figure 8, yes, it was already published but for use in this work we restructured only the figure and the information now presented in Table 3 to explain the meaning of each of the labels and which are the risk situations that are being quantified only through the helmet.

  1. Many of devices and results have been published before, I suggest the authors to reorganized the Introduction to show the real value of this paper: not proposing a new whole system, but making a promotion. It is necessary to show the new items and findings, not repeating the existing things. Which results support these new findings? How can we easily and clearly get them? They are all important for this paper.

Dear reviewer, thank you for your comments. It has been done as you, we have pointed out in the results that highlight this new perspective, specifically in:

“The ultimate goal is to have a complete platform with well-defined safety equipment for monitoring the strategic areas of the body, to reduce response times to accidents or problems that may occur during work that involves risk. That is, we will propose a way to handle the results of the previous devices with their respective AI/ML models, currently it is well known the method of ensemble models, Adabooks, Stacking ensemble ML, where our proposal is to perform the stacking of ML models and Deep models by means of the representation of the information as a complex graph. This allows us to handle information with non-linear behavior in a natural way, where, in the end, we are obtaining a voting of the information as any other ensemble model would do.”

  1. The information in Figs.12 and 14 is not well shown. What can we find from these figures?

Dear Dr., we thank you for your valuable contributions, we have improved the quality of the images and changed the colors when necessary, some of them were even recreated. Also, the information contained in them has been restructured. Which included information such as the following:

“…The network generated with the different outputs of the 4 integrated models, each colour represents the belonging to a certain model, for example, the 3 orange nodes represent the output of the LSTM (3 classes), the 4 blue nodes the 4 of the GMM, the 12 green nodes to the CNN and the rest to the Naive Bayes. This is how you have the joint responses of all the models.”

“Starting from a Graph we choose a set of heuristics available in the state of the art for complex networks (the most common ones are: Jaccard coefficient, Hub Promoted, Adamic Adar, etc.).

Referring to Jaccard Coefficient this allows us to establish a coefficient for a specific node (defined as similarity based on the neighborhood of the node), this value will become a characteristic for the input vector that will be used for the final modeling.

The process is repeated by assigning a value by heuristic to each node pair combination in the network, where at the end we will have an input vector for each node pair with a target that is identified with 0 or 1 depending on whether or not there is a link between each node pair.

Finally, an ML model is applied to the generated data where what we are doing is to weight the response of the different artificial intelligence models represented as a complex network to reduce the classification noise in industrial and work environments. The entire process is summarized in Figure”

  1. The Abstract and Conclusion should be written to show the new findings of this paper.

The scope of the work has been reflected and explained in the abstract and was improved the conclusion by adding all the new findings integrated into the paper. 

Round 3

Reviewer 1 Report

The latest submitted version has major improvements. The work is now easier to understand and better organized.

In declaring the use of Edge computing technologies, I would suggest to the authors that perhaps it would be appropriate to specify that the developed implementation falls within the specific Edge scope of Fog Computing.

Author Response

Thank you very much for the suggestion made by the reviewer. I think it is a very accurate comment and based on it: The selected architecture has been specified in more detail. The processing implementing artificial intelligence algorithms is carried out on a node above the devices. So as the reviewer expresses, it could be considered Fog Computing.

Reviewer 3 Report

The additional work is good. It can be accepted in my opinion after some minor revisions.

  1. It is better to give a citation for Figure 8 if its numbers have been published before.
  2. Add the website of Fig.1 in the reference list.

     3. The numbers of references is not in order. 8-10 in Line 72, then 29-32 in Line 79.  

Author Response

Thank you very much for the suggestions made by the reviewer.

It is better to give a citation for Figure 8 if its numbers have been published before.

We have added the reference to the figure

Add the website of Fig.1 in the reference list.

We have added to the reference list the website.

The numbers of references is not in order. 8-10 in Line 72, then 29-32 in Line 79.  

The notes made by the reviewer have been changed.
